# VRAG-RL: Empower Vision-Perception-Based RAG for Visually Rich Information Understanding via Iterative Reasoning with Reinforcement Learning

**Qiuchen Wang[1], Ruixue Ding, Yu Zeng[1], Zehui Chen[1], Lin Chen[1],**
**Shihang Wang, Pengjun Xie, Fei Huang, Feng Zhao[1]***
[1]MoE Key Laboratory of Brain-inspired Intelligent Perception and Cognition, USTC
qiuchenwang@mail.ustc.edu.cn, fzhao956@ustc.edu.cn

## Abstract

Effectively retrieving, reasoning and understanding visually rich information remains a challenge for traditional Retrieval-Augmented Generation (RAG) methods. On the one hand, traditional text-based methods cannot handle visual-related information. On the other hand, current vision-based RAG approaches are often limited by fixed pipelines and frequently struggle to reason effectively due to the insufficient activation of the fundamental capabilities of models. As reinforcement learning (RL) has been proven to be beneficial for model reasoning, we introduce **VRAG-RL**, a novel RL framework tailored for complex reasoning across visually rich information. With this framework, VLMs interact with search engines, autonomously sampling single-turn or multi-turn reasoning trajectories with the help of visual perception tokens and undergoing continual optimization based on these samples. Our approach highlights key limitations of RL in RAG domains: (i) Prior Multi-modal RAG approaches tend to merely incorporate images into the context, leading to insufficient reasoning token allocation and neglecting visual-specific perception; and (ii) When models interact with search engines, their queries often fail to retrieve relevant information due to the inability to articulate requirements, thereby leading to suboptimal performance. To address these challenges, we define an action space tailored for visually rich inputs, with actions including cropping and scaling, allowing the model to gather information from a coarse-to-fine perspective. Furthermore, to bridge the gap between users' original inquiries and the retriever, we employ a simple yet effective reward that integrates query rewriting and retrieval performance with a model-based reward. Our VRAG-RL optimizes VLMs for RAG tasks using specially designed RL strategies, aligning the model with real-world applications. Extensive experiments on diverse and challenging benchmarks show that our VRAG-RL outperforms existing methods by 20% (Qwen2.5-VL-7B) and 30% (Qwen2.5-VL-3B), demonstrating the effectiveness of our approach. The code is available at https://github.com/Alibaba-NLP/VRAG.

## 1 Introduction

Retrieval-Augmented Generation (RAG) [12, 18, 4] enables Language Models (LMs) to leverage external information to tackle problems. Due to the limitations of traditional textual RAG methods in handling visually rich information , efforts have been made to introduce RAG into the visual domain by integrating Vision-Language Models (VLMs) [1, 7, 33, 15, 34] with search engines. However, current visual RAG methods still fall short in effectively reasoning with search engines and

---

*Corresponding author

39th Conference on Neural Information Processing Systems (NeurIPS 2025).

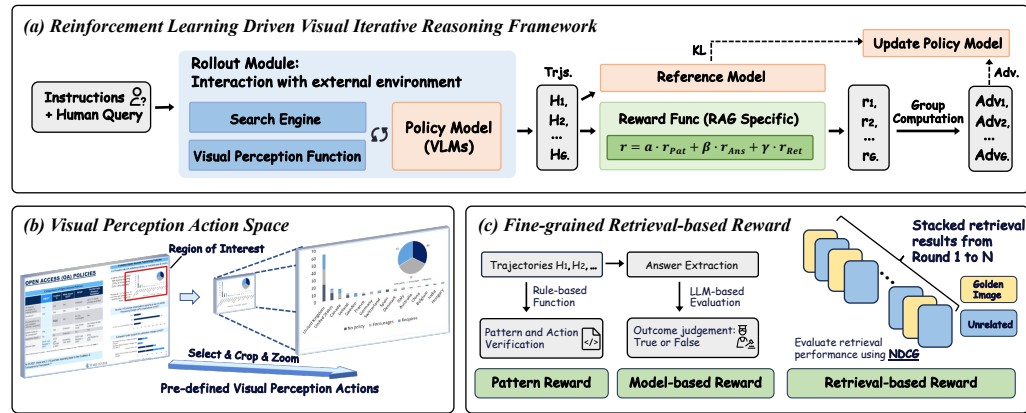

Figure 1: **Overall Framework of our Reinforcement Learning Framework.** (a) demonstrates the interaction process between the model and the external environment, as well as the implementation of the GRPO algorithm. (b) shows the proposed visual perception action space which allows the model to extract information from a coarse-to-fine perspective. (c) is the specially designed reward for RAG, which combines outcome and retrieval performance across the entire sampling process.

understanding complex visual information. Reinforcement Learning (RL) has been recognized as an effective approach for optimizing VLMs in complex reasoning tasks [39, 20, 14, 30, 49]. Therefore, RL offers a promising approach to address the challenges faced by visual RAG methods.

Inspired by these advancements, we introduce **VRAG-RL**, a novel multimodal RL framework specifically designed for iterative reasoning in visually rich information RAG. Our approach is based on three critical observations: **(i) Insufficient activation of reasoning capabilities with visual information.** Existing methods underutilize the reasoning potential of VLMs when incorporating visual information. For instance, prior approaches tend to merely embed images into the context without adequately addressing visual-specific perception processes, resulting in insufficient reasoning token allocation and limiting the models' ability to fully leverage visual data for complex reasoning tasks. **(ii) Inefficient and disjointed Retrieval.** In previous work, limited by the inability to articulate complex requirements, models struggled to retrieve relevant information efficiently, which may lead to repetitive and meaningless interactions, restricting the overall effectiveness. **(iii) Inconsistent multi-turn reasoning and unstable training with VLMs.** Current RL frameworks for LMs often struggle with maintaining stability and consistency during multi-turn reasoning. Handling complex, multi-step reasoning tasks can be particularly challenging, as models may encounter difficulties in maintaining effective reasoning across interactions with external environments, leading to inconsistent performance and suboptimal results. This challenge is further exacerbated for VLMs, which are limited by their instruction-following and reasoning capabilities.

Building upon these insights, VRAG-RL introduces improvements in various modules: (i) We propose a visual perception action space that includes selecting regions of interest and zooming into these areas. VLMs with visual perception tokens in the action space are capable of acquiring information from coarse-to-fine perspective. As shown in Figure 1 (b), when dealing with images or charts within documents, VLMs can give higher attention to information-dense areas through the proposed perception tokens. This allows the model to more effectively activate reasoning abilities within a limited context length, preventing the overlooking of details. (ii) Furthermore, rather than relying solely on a simple outcome-based reward, we factor in the effectiveness of the retrieval process as part of the reward structure. In particular, during the interaction between the model and the search engine, retrieving pertinent images promptly enhances the model's ability to address questions effectively, whereas persistently retrieving irrelevant documents adds noise and hampers the reasoning process. As illustrated in Figure 1 (c), by integrating retrieval performance into reward, we establish comprehensive guidance for retrieval-augmented generation frameworks. (iii) Inspired by the current think-then-answer approach and the ReAct paradigm, we model the interaction between the VLMs and the search engine, along with the visual perception action space, as a process of iterative reasoning and tool invocation. Figure 1 (a) illustrates our training pipeline, which supports automatic sampling and integrates the GRPO algorithm. To ensure stability in multi-turn sampling

and training, we have carefully designed the sampling strategy including post-processing for each interaction, and model-based reward together with the retrieval reward mentioned above guides the model training. Additionally, we have re-annotated existing datasets of visually rich documents and developed a data construction pipeline to efficiently scale data for RL and SFT.

Our major contributions are as follows:

- We propose VRAG-RL, a novel reinforcement learning framework tailored for training VLMs to effectively reason, retrieve, and understand visually rich information.
- We define a visual perception action space that includes selecting, cropping, and scaling regions of interest, allowing VLMs to gather information progressively from coarse-grained to fine-grained levels. This action space enhances the models' ability to focus on information-dense areas and activates their vision-specific reasoning capabilities more effectively.
- We introduce a comprehensive reward structure that integrates retrieval performance and model-based outcome reward. This reward mechanism aligns the model more closely with real-world applications, bridging the gap between users' original intentions and the retriever.
- Extensive experiments demonstrate the effectiveness of our method. VRAG-RL significantly outperforms strong baselines, achieving over 20% improvement on various benchmarks.

## 2 VRAG-RL

In this section, drawing on insights and foundational ideas, we present a comprehensive description of our **VRAG-RL** framework. We start with the formulation of the problem (§2.1), then introduce the action space designed for visual perception (§2.2) and the fine-grained reward specifically defined for the RAG task (§2.3). Finally, we illustrate the model interaction process in the rollout module and the reinforcement learning training implementation of our framework (§2.4).

### 2.1 Problem Formulation

Given a query denoted as $q$, we have a huge collection of images $\mathcal{C} = \{\mathbf{I}_1, \mathbf{I}_2, \ldots, \mathbf{I}_N\}$, consisting of $N$ images. Each image contains a variety of visually rich elements, such as flowcharts, charts, tables, and diverse layouts, derived from real-world documents across multiple domains, including slides and reports. Our goal is to efficiently reason, accurately retrieve the most relevant images, extract valuable information from the complex visual data, and generate the final answer $a$ to the query $q$.

### 2.2 Visual Perception Action Integration for Understanding Information-Dense Regions

Previous works merely involved migrating textual RAG to the multi-modal domain, which simply meant inserting images into the context and then reasoning and responding. However, these efforts overlooked the characteristics of image data, where the efficiency of visual perception is closely related to image resolution, visual element layouts, information density, and other visually related factors. Motivated by these findings, we introduce a dynamic novel visual perception paradigm into VLMs that involves region selection and re-encoding at the token level, as illustrated in Figure 2.

**Definition of Visual Perception Actions.** We define the visual perception action space for VLMs by taking into account the specific characteristics of visual information. This enables the model to select regions with high information density or regions relevant to the query for a detailed view, acquiring information from a coarse to fine perspective. We integrate search queries, answer summaries, and visually specific actions into a unified action space to align with the model's pre-training domain.

The policy model $\pi_\theta$ interacts with the environment in the Thought-Action-Observation $(\mathcal{T}, \mathcal{A}, \mathcal{O})$ paradigm. In each interaction, the model generates the next action $\mathcal{A}_t \sim \pi_\theta(\cdot \mid \mathcal{H}_{t-1})$ based on the trajectory $\mathcal{H}_{t-1}$ from step $t-1$ and earlier. A role-based function is used to extract visual perception tokens <region> and </region>, whose main purpose is to select, crop, and zoom in on the region of interest of the image that has already been retrieved in the context:

$$\mathcal{A}_t \times \mathcal{O}_k \to \mathcal{O}_t, k \in \{1, 2, \ldots, t-1\}, \tag{1}$$

Given a $w \times h$ image as an observation $\mathcal{O}_k$, a bounding box $[x_{min}, y_{min}, x_{max}, y_{max}]$ within perception tokens can precisely delineate the position of region $\mathcal{R}$, where $(x_{min}, y_{min})$ and $(x_{max}, y_{max})$

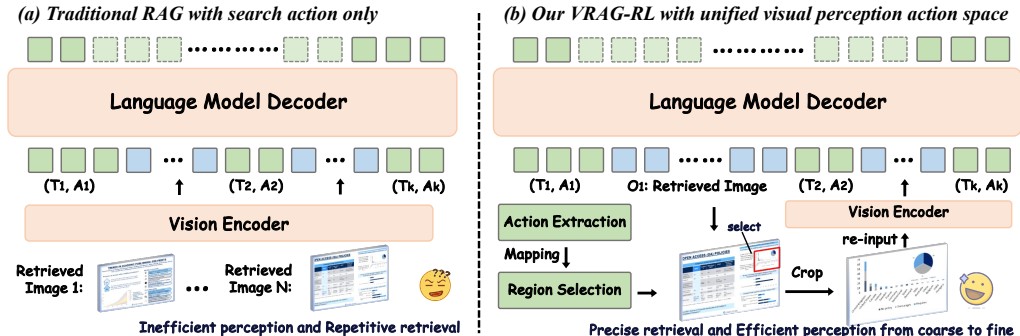

Figure 2: **Comparison between our VRAG-RL and the traditional RAG in terms of perception methods.** (a) Traditional methods lack effective perception, which easily leads to repetitive and ineffective retrieval calls and suboptimal outcomes. (b) Our VRAG-RL is efficient and accurate, enabling the model to perceive information-dense regions from a coarse-to-fine perspective.

represent the coordinates of the top-left and bottom-right pixels of region $\mathcal{R}$. Some current models' pre-training domains for grounding tasks normalize the coordinates to $[0, \delta]$, resulting in actual coordinates of $(x \times \frac{w}{\delta}, y \times \frac{h}{\delta})$, while other models, such as Qwen2.5VL, directly use the original coordinates without normalization. Then we will map the selected region $\mathcal{R}$ from the image tokens in context to the $w_{raw} \times h_{raw}$ raw image, and crop this raw image to obtain $\hat{\mathcal{R}}$:

$$\hat{\mathcal{R}} = Crop(\mathbf{I}_{raw}, [x_{min} \times \frac{w_{raw}}{w_{encoder}}, y_{min} \times \frac{h_{raw}}{h_{encoder}}, x_{max} \times \frac{w_{raw}}{w_{encoder}}, y_{max} \times \frac{h_{raw}}{h_{encoder}}]). \quad (2)$$

where $(w_{raw}, h_{raw})$ are the shape of the original image $\mathbf{I}_{raw}$, $(w_{encoder}, h_{encoder})$ are determined by the vision encoder such that $w_{encoder} \times h_{encoder} = Pixels_{max}$. Finally, $\hat{\mathcal{R}}$ is integrated into the context as an observation: $\hat{\mathcal{R}} \to \mathcal{O}_t$. Actually, the image token embedded in the context does not represent the original size of the image. The maximum pixel size $Pixels_{max}$ for the vision encoder is often considerably smaller than the pixel of visually rich documents found in real-world applications. This is the reason why the region cropped from the original image and scaled within the vision encoder has a higher density of vision tokens. This simple yet effective "crop and re-input" strategy enhances visual perception performance by directly increasing perceptual resolution [50, 26, 37].

**Trajectory Data Scaling-Up Based on Multi-Expert Sampling.** To effectively train the model, especially smaller-scale models, to learn the utilization of Visual Perception Tokens while retaining their foundational capabilities, we need to train them with high-quality data through Supervised Fine-Tuning before applying RL. We propose a multi-expert sampling strategy to scale up the trajectory data, aiming to sample diverse interactions within the same reasoning trajectory for each data.

The core idea is to utilize large-scale models $\pi_{LM}$ to effectively guide the reasoning process and tool selections within a trajectory, while smaller expert models $\pi_{EM}$ annotate coordinate under the guidance of large-scale models. At the $t_{th}$ interaction between the model and the environment:

$$\mathcal{H}_t = \{\mathcal{T}_1, \mathcal{A}_1, \mathcal{O}_1, \cdots, \mathcal{O}_{t-1}, \mathcal{T}_t, \mathcal{A}_t, \mathcal{O}_t\}, \quad (3)$$

where $\mathcal{H}_t$ is the trajectory, representing the sequence of past observations and actions leading up to the current step. The $\pi_{LM}$ equipped with extensive capacities for understanding and processing complex multi-modal interactions, act as pioneers in determining the overarching reasoning pathway:

$$\{\mathcal{T}_t, \mathcal{A}_t\} = \pi_{LM}(\cdot \mid \mathcal{H}_{t-1}), \quad (4)$$

We use a rule-based function to extract action and thought. If the action is search, the engine returns the original image as $\mathcal{O}_t$. Otherwise, each time a visual perception token is output, we employ grounding-specific expert models to re-locate the coordinates of regions of interest:

$$\hat{\mathcal{A}}_t = \pi_{EM}(\cdot \mid \mathcal{H}_{t-1}; \mathcal{T}_t), \quad (5)$$

where the expert models $\pi_{EM}$ benefit from the guidance provided by the large model's thought $\mathcal{T}_t$, leveraging these insights to enhance their precision in region localization. The newly generated

coordinates of the region of interest $\hat{\mathcal{A}}_t$ will replace the old visual perception tokens $\mathcal{A}_t$ generated by $\pi_{LM}$, and the re-encoded image serves as observation $\hat{\mathcal{O}}_t$:

$$\hat{\mathcal{O}}_t = \mathcal{P}_V(\mathcal{O}_{t-1}, \hat{\mathcal{A}}_t). \tag{6}$$

where $\mathcal{P}_V$ represents the visual processing function, the selected region will undergo cropping, zooming in, and re-encoding before being inserted into the context.

### 2.3 Fine-Grained Reward Function Tailored for Enhancing RAG Framework

Unlike traditional RL methods that focus only on output results, VRAG-RL emphasizes optimizing retrieval in RAG, as retrieval quality directly affects overall performance. We designed a reward function with three components: pattern reward, retrieval efficiency reward, and model-based outcome reward, guiding the model to efficiently retrieve information and generate high-quality answers.

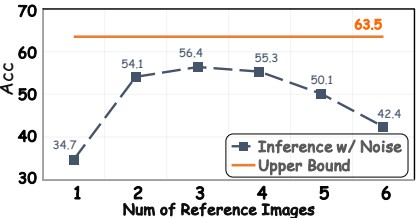

Figure 3: Experiments on the impact of context length on model performance.

**Retrieval Efficiency Reward.** As shown in Figure 3, when the information is sufficient, an excessively long context can interfere with the model. Therefore, the earlier and more comprehensive the retrieval of relevant information, the better the model can construct a coherent and informative context for generating high-quality answers. Inspired by Normalized Discounted Cumulative Gain, and using our predefined relevance of the recalled images, we define:

$$\text{DCG}(\mathcal{D}_{trj}) = \sum_{i=1}^{|\mathcal{D}_{trj}|} \frac{2^{s_i} - 1}{\log_2(i+1)}, \quad s_i = \begin{cases} 1, & \text{if } d_i \in \mathcal{D}_{rel} \\ 0, & \text{if } d_i \notin \mathcal{D}_{rel} \end{cases}, \tag{7}$$

where $d_i \in \mathcal{D}_{trj}$ represents stacked retrieved images within the trajectory, $\mathcal{D}_{rel}$ is the collection of relevant golden images, $s_i$ is the predefined relevance score. We believe that the performance is optimal when all relevant documents are retrieved first, the Ideal-DCG is defined as:

$$\text{IDCG}(\mathcal{D}_{rel}) = \sum_{i=1}^{|\mathcal{D}_{rel}|} \frac{2^{s_{\text{rel}}} - 1}{\log_2(i+1)} + \sum_{i=|\mathcal{D}_{rel}|+1}^{n} \frac{2^{s_{\text{unrel}}} - 1}{\log_2(i+1)} = \sum_{i=1}^{|\mathcal{D}_{rel}|} \frac{1}{\log_2(i+1)}, \tag{8}$$

where $s_{\text{rel}} = 1$ and $s_{\text{unrel}} = 0$ respectively represent the relevance scores of ideally relevant and irrelevant documents. Our Retrieval Efficiency Reward is defined as:

$$r_{Ret} = \frac{\text{DCG}(\mathcal{D}_{trj}, \mathcal{D}_{rel})}{\text{IDCG}(\mathcal{D}_{rel})}. \tag{9}$$

where $r_{Ret}$, the modified NDCG, is directly used as the reward to reflect retrieval performance.

**Pattern Consistency and Model-Based Outcome Rewards.** The rule-based pattern reward is designed to encourage the model to follow the reasoning patterns during the interaction process:

$$r_{\text{Pat}} \sim Parse(\mathcal{H}), \tag{10}$$

where $\mathcal{H}$ is the generated trajectory. $Parse(\cdot)$ employ action tokens `<search>` and `</search>` to extract predefined actions in the action space. This is crucial for a reasoning agent with a predefined action space, as it helps effectively extract actions and thoughts. Regarding outcome reward, unlike rule-based methods that are prone to falling into local optima, we adopt a model-based reward:

$$r_{\text{Ans}} \sim \pi_{\text{RM}}(\cdot | \mathcal{Q}, \mathcal{A}_{\text{golden}}, \mathcal{A}_{\text{pred}}), \tag{11}$$

where $\mathcal{Q}$ represents the input query, $\mathcal{A}_{\text{golden}}$ is the reference golden answer, and $\mathcal{A}_{\text{pred}}$ is the answer generated by the VLMs. Based on these inputs, the evaluation model $\pi_{\text{RM}}$ assesses the correctness of the final answer. Please refer to Appendix A for the detailed prompt used in the model-based reward.

**Integrated Reward Function.** The final reward function is a weighted combination of the three components described above, with weights used to balance the contributions of each component:

$$r_\phi = \alpha \cdot r_{Ret} + \beta \cdot r_{Ans} + \gamma \cdot r_{Pat}. \tag{12}$$

where $\alpha + \beta + \gamma = 1$. In practice, we usually set $\gamma = 0$ as the model can effectively learn the pattern after SFT. We set $\gamma = 0.1$ when performing RL with cold start to help the model learn the predefined pattern. By integrating these three components into the reward function, our VRAG-RL provides a comprehensive and fine-grained evaluation mechanism that guides the model in optimizing its reasoning and retrieval capabilities in a way that aligns closely with real-world applications.

## 2.4 Reinforcement Learning Framework with Iterative Reasoning

We apply RL to multimodal RAG agent tasks to enhance the capability of VLMs in retrieving and reasoning. Our RL framework is primarily divided into two parts for discussion: the rollout process for multimodal agent and the reinforcement learning training strategy for multi-turn interactions.

**Multi-Round Generation with Search Engine and Visual Perception Actions.** As shown in Algorithm 1, the model interacts with the external environment in multiple turns, where the observation, which is the image, is inserted into the trajectory in the role of the user. This is necessary to align with the model's pre-training domain, where only the user token can insert image tokens.

---

**Algorithm 1** Interaction of VLM with the External Environment through Iterative Reasoning

---

**Input:** Input query $x$, Policy model $\pi_\theta$, External environment $\mathcal{V}$, Maximum iterations $T$.
**Output:** Final trajectory $y$.
1: Initialize rollout sequence $y \leftarrow \emptyset$ and action count $t \leftarrow 0$
2: **while** $t < T$ **do**
3:     Generate VLM response sequence $y_t \sim \pi_\theta(\cdot \mid x, y)$
4:     Concatenate $y_t$ to the $y$ sequence with the role of assistant: $y \leftarrow y + y_t$
5:     **if** `<search> </search>` detected in $y_t$ **then**
6:         Extract search query $q \leftarrow Parse(y_t)$ and Retrieve related image $I_t = Ret(q)$
7:     **else if** `<region> </region>` detected in $y_t$ **then**
8:         Extract visual perception tokens $loc \leftarrow Parse(y_t)$ and Processing image $I_t = P_V(loc, y)$
9:     **else if** `<answer> </answer>` detected in $y_t$ **then**
10:        **return** final generated trajectory $y$
11:     **end if**
12:     Concatenate vision tokens $I_t$ to the sequence $y$ with the role of user: $y \leftarrow y + I_t$
13:     Increment action count $t \leftarrow t + 1$
14: **end while**
15: **return** final generated trajectory $y$

---

**Training Strategy for Reinforcement Learning in Multi-Step Interactions.** We propose a RL framework that enables VLM to learn how to interact with search engines and gather visually rich information from a coarse-to-fine perspective. The optimization objective is formulated as:

$$\max_{\pi_\theta} \mathbb{E}_{x \sim \mathcal{D}, y \sim \pi_\theta(\cdot \mid x; \mathcal{V})} \left[ r_\phi(x, y) \right] - \beta \mathbb{D}_{\mathrm{KL}} \left[ \pi_\theta(y \mid x; \mathcal{V}) \,\|\, \pi_{\mathrm{ref}}(y \mid x; \mathcal{V}) \right], \tag{13}$$

where the $\pi_\theta$ is the policy model, $\pi_{ref}$ is the reference model, $\mathbb{D}_{\mathrm{KL}}$ is KL-divergence, and $y \sim \pi_\theta(\cdot \mid x; \mathcal{V}) = \pi_\theta(\cdot \mid x) \otimes \mathcal{V}$ is the rollout process. Our approach implements Group Relative Policy Optimization (GRPO) [13], which optimizes the model's retrieval-augmented reasoning capability with group-sampled role-play trajectories. Please refer to Appendix C for more details.

## 3 Experiments

### 3.1 Experimental Settings

**Datasets, Metric and Baselines.** To evaluate the effectiveness of VRAG-RL, we compare our method with the text-based and vision-based baselines: (1) **Vanilla RAG** [11] uses the original question as a query for the search engine, then VLMs perform direct inference. (2) **ReAct** [48]: The model performs rewriting, retrieving, and reasoning in the think-then-act paradigm. (3) **Search-R1(-VL)** is the baseline adapted from Search-R1 [19], and the settings are aligned across all experiments

Table 1: **Main Results.** The best performance are marked in bold. SlideVQA and ViDoSeek mainly focus on reasoning type, while MMLongBench focuses on the visual type of reference content. OCR-based (�) RAG and purely visual (◉) RAG are evaluated with the same prompt and setting.

| METHOD | SLIDEVQA | | VIDOSEEK | | MMLONGBENCH | | | | | OVERALL |
|---|---|---|---|---|---|---|---|---|---|---|
| | Single-hop | Multi-hop | Extraction | Logic | Text | Table | Chart | Figure | Layout | |
| *Qwen2.5-VL-3B-Instruct* | | | | | | | | | | |
| ✍ Vanilla RAG | 15.1 | 12.1 | 8.8 | 14.3 | 3.9 | 5.1 | 1.7 | 3.1 | 2.5 | 11.2 |
| ✍ ReAct [48] | 11.8 | 9.9 | 5.3 | 7.4 | 6.5 | 3.7 | 3.9 | 5.2 | 2.5 | 8.4 |
| ✍ Search-R1 [19] | 17.5 | 13.8 | 13.3 | 20.7 | 3.4 | 3.2 | 4.5 | 4.1 | 6.8 | 14.1 |
| ◉ Vanilla RAG | 19.4 | 12.2 | 10.1 | 17.3 | 2.2 | 4.1 | 5.2 | 4.7 | 4.3 | 13.2 |
| ◉ ReAct [48] | 15.7 | 10.9 | 6.7 | 14.2 | 2.7 | 3.6 | 3.4 | 3.1 | 5.1 | 10.9 |
| ◉ Search-R1-VL [19] | 26.3 | 20.1 | 20.1 | 29.8 | 8.5 | 7.8 | 7.9 | 9.3 | 7.6 | 21.3 |
| ◉ **VRAG-RL (Ours)** | **65.3** | **38.6** | **63.1** | **73.8** | **22.7** | **16.1** | **21.9** | **21.4** | **19.5** | **53.5** |
| *Qwen2.5-VL-7B-Instruct* | | | | | | | | | | |
| ✍ Vanilla RAG | 26.1 | 10.6 | 24.7 | 30.9 | 8.5 | 5.4 | 11.7 | 4.4 | 3.3 | 20.9 |
| ✍ ReAct [48] | 21.2 | 13.3 | 14.3 | 21.3 | 5.9 | 5.1 | 7.3 | 5.5 | 1.7 | 15.8 |
| ✍ Search-R1 [19] | 28.4 | 19.7 | 20.8 | 30.6 | 9.9 | 6.0 | 7.9 | 10.1 | 5.9 | 22.2 |
| ◉ Vanilla RAG | 29.1 | 17.4 | 26.4 | 41.3 | 13.1 | 14.7 | 15.9 | 4.3 | 7.6 | 24.2 |
| ◉ ReAct [48] | 34.8 | 20.4 | 27.5 | 42.1 | 10.1 | 12.4 | 10.2 | 6.2 | 7.1 | 26.9 |
| ◉ Search-R1-VL [19] | 48.3 | 42.3 | 40.5 | 50.3 | 19.9 | 13.4 | 12.9 | 11.4 | 10.2 | 37.4 |
| ◉ **VRAG-RL (Ours)** | **69.3** | **43.1** | **60.6** | **74.8** | **26.1** | **26.3** | **24.8** | **25.9** | **21.2** | **57.1** |

to ensure fairness. We evaluate our method on three challenging, visually rich benchmarks: **ViDoSeek** [41], **SlideVQA** [40] and **MMLongBench** [29]. The model-based evaluation metric is binary 0 or 1, indicating the accuracy of the model's responses. Please refer to Appendix E and F for more details.

**Training and Inference Setups.** We conducted SFT and RL on llama-factory [53] and verl [38] respectively. We use full parameter fine-tuning and cosine learning scheduler with a warmup ratio of 0.1 during SFT. When training with the GRPO algorithm, we set the group size to 5 and the coefficient for the KL loss is typically set to 0.01, but if we perform cold start, we set it to 0 to disable the KL loss constraint on the model. During training and inference, we built a search engine from a database of approximately ∼ 70k visual documents. All the experiments are conducted on 8 NVIDIA A100 80G GPUs. Please refer to Appendix G for detailed hyperparameters used in our paper.

## 3.2 Results

**Main Results.** As shown in Table 1, compared to purely visual methods, OCR-based methods exhibit significant limitations on visually intensive benchmarks. On the one hand, visual information inherently contains elements that cannot be represented by text, such as element positions, layout, and color, etc. On the other hand, the perceptual capabilities of OCR models are considerably inferior to those of the current advanced VLMs, which restricts the overall performance ceiling of the framework. Visual-based methods have proven to be a more elegant solution compared to OCR-based methods, especially in tasks related to visual understanding. For prompt-based baselines of vision domain, Vanilla RAG and ReAct exhibit poor performance, far behind RL-based baselines and our method on various benchmarks. The 7B model, compared to the 3B model, possesses superior perception and understanding capabilities, exhibiting strong performance across various datasets. For RL-based baselines, our method also performs better than search-R1-VL on both Qwen2.5-VL-7B-Instruct ($34.7 \rightarrow 57.1$) and Qwen2.5-VL-3B-Instruct ($21.3 \rightarrow 53.5$). The evaluation results on SlideVQA and ViDoSeek demonstrate our model's significant improvement in reasoning capabilities across various reasoning tasks. Furthermore, as MMLongBench includes multiple visual elements, which indicates the model's improvement in visual perception capabilities, this phenomenon is related to our proposed visual perception action space. The results across various benchmarks prove the effectiveness and generalization of our method in the retrieval and reasoning of visually rich information.

**Approach Ablations.** As shown in Table 2, taking Qwen2.5-VL-7B-Instruct as an example, we decompose the key components of VRAG-RL to examine the impact of different rewards and action space on performance separately. In a macro view, removing each module results in a clear drop in the accuracy, which validates the power of our RAG-specific reward and Visual-perception action space.

The action space module we defined shows a certain degree of improvement on different bases, which **proves the effectiveness of the visual perception-based strategy**. Consistent with the findings demonstrated in MMLongBench in Figure 5, the visual perception action space we introduced has generally enhanced the frame-

Table 2: Ablation study on three benchmarks.

| REWARD | | ACTION SPACE | | Accuracy |
|---|---|---|---|---|
| Vanilla | RAG-Specific | Search | Visual-Perception | |
| ✓ | | ✓ | | 47.2 |
| ✓ | | ✓ | ✓ | 49.3 |
| | ✓ | ✓ | | 54.9 |
| | ✓ | ✓ | ✓ | 57.1 |

work's performance, particularly in improving high-density visual information. Furthermore, ablation experiments on the reward model further demonstrate that retrieving relevant information is a prerequisite for high-quality generation, highlighting the role of high-quality retrieval in RAG, which **proves the importance of our RAG-specific reward**. Comparisons and analyses of experiments across different settings collectively demonstrate the effectiveness and generalization of our modules, and their combination comprehensively enhances end-to-end performance from various perspectives.

### 3.3 Analysis

**Better retrieval facilitates high-quality generation.** Our VRAG-RL framework significantly enhances the retrieval efficiency, which is crucial for constructing a coherent and informative context for high-quality generation. As demonstrated in Figure 3, the context length has a substantial impact on model performance. When the context is too long, it can introduce noise and interfere with the model's ability to generate accurate answers. In contrast, when relevant information is retrieved early and comprehensively, the model can build a more focused and informative context. As shown in Figure 4, our model is more effective at

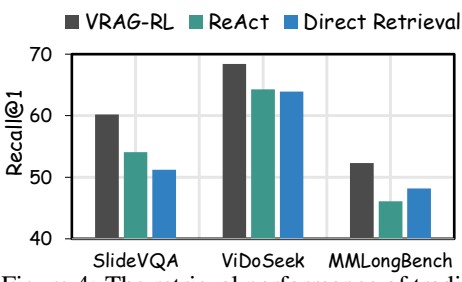

Figure 4: The retrieval performance of traditional prompt-based RAG and our approach.

retrieving relevant information compared to traditional prompt-based rewrite methods. Our approach provides the vision model with a better context for generating high-quality answers.

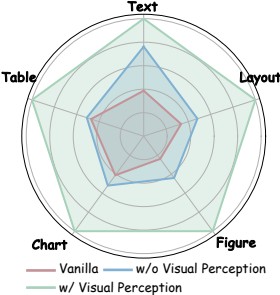

Figure 5: Relative performance on MMLongBench.

**Visual perception action space provides a fine-grained perspective.** The visual perception action space introduced in our framework further enhances understanding by allowing the model to focus on information-dense regions of images. Figure 5 illustrates the relative performance comparison between our approach with visual perception action space and various baselines, from which we can observe that VRAG-RL not only performs well in textual tasks but also shows noticeable improvements in tasks requiring visual perception abilities, particularly in Layout, Chart, and Figure. This is particularly important given the current limitations in computational resources, especially considering that VLMs are highly memory-intensive. Using this dynamic resolution strategy, the model can achieve more detailed perception within the constraints of limited computational resources, rather than simply maximizing the resolution of the original image. Our method achieves an improvement in perceptual abilities while optimizing resource utilization. Perhaps this human-like way of thinking and acting is the key to AGI.

**Reinforcement learning helps the model to perform multi-step reasoning effectively.** One major challenge of the prompt-based method is that as the number of interactions increases, the model's capability to follow instructions weakens. However, pre-training with SFT helps the model reason in a pre-defined pattern compared to cold start, but it also impacts

Table 3: Average Finish Rate (%) and Average Invalid Action Rate (%).

| Method | Invalid Action Rate ↓ | Finish Rate ↑ |
|---|---|---|
| SFT | 9.4 | 84.2 |
| + RL | **5.1** | **97.1** |

the model's inherent foundational capabilities to some extent. To further explore the activation of multi-turn reasoning abilities in models by RL, we compared the iterative reasoning performance of models with and without RL, as shown in Table 3. For our method with action space, effective actions are crucial for interacting with the external environment. The Invalid Action Rate indicates incorrect action responses, which include not only pattern errors but also hallucinations caused by wrong cropping, answering before retrieval, and so on. Inefficient reasoning often includes repeated

meaningless searches, leading to a decrease in the finish rate. Our method with RL effectively reduces the invalid rate and increases the finish rate. It guides the model to make optimal decisions at each step of the reasoning process, enabling it to flexibly adjust strategies when faced with different types of out-of-domain visual information, thereby better completing complex reasoning tasks.

**Model-based reward offers more stable training compared to rule-based reward.** Previous works often use EM as the reward, which is too strict. Unlike short answers for data-related questions, it is difficult for the model's responses to exactly match the golden answer, resulting in inefficient training. However, using recall as a reward may lead to misjudgments and cause models to hack the function, resulting in repetitive responses that destabilize training. In contrast, a model-based reward leverages an evaluation model to assess the quality and relevance of generated responses in a more flexible manner. This approach not only aligns better with real-world applications but also provides a more stable and effective training signal, as demonstrated in Appendix A. The model-based reward thus enables VRAG-RL to achieve more robust performance across visual reasoning tasks.

**Time efficiency.** As shown in Figure 6, our method's multi-turn interaction with external environments can lead to increased latency. The latency of vanilla RAG remains consistent, as it only performs a single search and provides an answer. ReAct RAG, a prompt-based method, also demonstrates multi-turn interaction capabilities due to the fundamental reasoning abilities of the model. However, it is limited

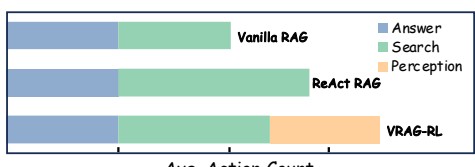

Figure 6: Latency Analysis on Generation.

to only two defined actions: answer and search. Due to the lack of sufficient perception capabilities, it often falls into repetitive search loops. Our approach equips the model with a visual perception space that can effectively understand visually rich images. The model can quickly extract answers after retrieval, thus avoiding ineffective searches. Despite the increase in latency, the overall performance improves due to the higher quality of generated answers, making the trade-off between latency and accuracy highly beneficial for visually rich retrieval and understanding tasks.

**Case Study.** In Figure 7 and 8 (Appendix H), we list the trajectories of our VRAG-RL to illustrate how our model reasons and interacts with the environment. These cases highlight two challenges in visually rich information RAG: (1) accurately retrieving relevant images, and (2) the reference information often requires higher-resolution perception. In Figure 7, we can observe that the model demonstrated reflective capability, and eventually identified subtle clues in the relevant images. Moreover, as shown in Figure 8, the model engages in visual perception actions only when required, showcasing human-like reasoning instead of simply replicating patterns from its training data.

## 4 Related Work

**Vision-based Retrieval-augmented Generation.** RAG demonstrates significant advantages in addressing knowledge-intensive problems [22, 12, 2]. Traditional text-based RAG methods typically involve designing different agents to interact with search engines [45, 5, 6, 44, 25, 32, 21, 10]. However, with the widespread adoption of electronic documents, knowledge is no longer confined to text. Recently, there has been an increasing amount of research on OCR-free retrieval methods that directly align textual queries with images [51, 11]. Furthermore, more and more work is focusing on multimodal RAG agents [41, 8, 16, 24, 46], enabling more accurate retrieval and extraction of visual information. Our work builds upon these developments by incorporating visual perception actions into visual-based RAG, effectively activating the reasoning and understanding capabilities of VLMs.

**Reinforcement Learning with Large Models.** Reasoning capabilities are crucial for models to effectively address complex problems, and RL has been proven to be a powerful approach to enhance these capabilities [13, 15]. Previous work applied RL in the training of LLMs [31, 43, 35, 36, 13, 27]. Additionally, more and more works aim to use RL to enhance the reasoning capabilities of VLMs [3, 30, 28, 52]. Recent advancements have seen RL being widely applied to the training of large model-driven agents [42]. These agents, especially RAG agents, require robust multi-step reasoning capabilities to interact effectively with external environments [17, 23]. However, there is still a scarcity of RL frameworks specifically tailored for multimodal iterative reasoning, which is essential

for handling visually rich information. Our work aims to fill this gap by introducing a novel RL framework that enables VLMs to perform iterative reasoning with visual perception actions, thereby enhancing their reasoning capabilities in complex, multi-modal retrieval-augmented reasoning tasks.

# 5   Conclusion and Future Work

In this paper, we introduce VRAG-RL, a novel reinforcement learning framework tailored for complex reasoning across visually rich information. Our approach enables Vision Language Models to interact with search engines more effectively, significantly enhancing their reasoning and retrieval capabilities. Extensive evaluations on various benchmarks have demonstrated significant advantages in visual information reasoning, retrieval, and understanding with our model. For future work, we plan to introduce more actions that mimic how humans handle complex information, allowing the model to focus more on deep thinking. Additionally, we aim to reduce hallucinations by leveraging more advanced models, further improving the accuracy and reliability of our framework.

## Acknowledgments

This work was supported by the Anhui Provincial Natural Science Foundation under Grant 2108085UD12. We acknowledge the support of GPU cluster built by MCC Lab of Information Science and Technology Institution, USTC. The AI-driven experiments, simulations and model training were performed on the robotic AI-Scientist platform of Chinese Academy of Sciences.

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

# Appendix

## A  Model-Based Reward

We employ a model-based reward to evaluate the quality and relevance of generated responses. Specifically, we utilize Qwen2.5-7B-Instruct [47] as our reward model. This model is deployed on 4 NVIDIA A100 GPUs to enable efficient batch evaluation. The prompt used for the reward model is illustrated in Figure 12. Given the input query, reference answer, and generated response, the reward model assesses the correctness of the generated response and outputs a binary value (0 or 1) to represent the accuracy of the answer. Compared to the rule-based reward like exact match (EM) or Recall, used in previous work [19, 3], our model-based reward provides a more flexible and comprehensive evaluation of the generated response. This leads to higher training efficiency and better generalization to diverse datasets.

## B  The implementation of the search engine

To effectively support the retrieval-augmented generation tasks in our VRAG-RL framework, we implemented OCR-based and vision-based pipeline separately. The vision-based retriever is built upon the state-of-the-art embedding model ColPali [11], which is specifically designed for aligning textual queries with images. For the textual retrieval pipeline, we employ the PP-OCR [9] to extract text from images. We utilize the Llama-Index to ensure an efficient indexing and querying mechanism for large-scale image datasets. In our experiments, we deployed the search engine on a single NVIDIA A100 80G GPU, allowing us to handle large-scale queries efficiently. The use of batch querying further optimizes the retrieval speed, making it suitable for real-time applications.

## C  Reinforcement Learning Framework with GRPO

Our framework implements the Group Relative Policy Optimization (GRPO), which leverages the average reward of multiple sampled outputs as a baseline rather than relying on a learned value function. The policy model is optimized by maximizing the following objective function:

$$\mathcal{J}_{GRPO}(\theta) = \mathbb{E}_{x \sim \mathcal{D}, \{y_i\}_{i=1}^G \sim \pi_{\text{old}}(\cdot|x;\mathcal{V})} \left[ \frac{1}{G} \sum_{i=1}^G \frac{1}{\sum_{t=1}^{|y_i|} I(y_{i,t})} \sum_{t=1:I(y_{i,t})=1}^{|y_i|} \min \left( \frac{\pi_\theta(y_{i,t}|x, y_{i,<t};\mathcal{V})}{\pi_{\text{old}}(y_{i,t}|x, y_{i,<t};\mathcal{V})} \hat{A}_{i,t}, \right. \right.$$

$$\left. \text{clip} \left( \frac{\pi_\theta(y_{i,t}|x, y_{i,<t};\mathcal{V})}{\pi_{\text{old}}(y_{i,t}|x, y_{i,<t};\mathcal{V})}, 1-\epsilon, 1+\epsilon \right) \hat{A}_{i,t} \right) - \beta \mathbb{D}_{KL} \left[ \pi_\theta || \pi_{\text{ref}} \right]$$

where rollout module samples a group of trajectories $\{y_1, y_2, \ldots, y_G\}$ from the reference policy $\pi_{\text{ref}}$ for each input question $x$ by interacting with the external environment $\mathcal{V}$. $\hat{A}_{i,t}$ represent the advantage, computed based on the relative rewards of outputs within each group.

## D  Expert Trajectories Collection

**Data Collection.**  To train our model effectively, we collected expert trajectories using Qwen-VL-max-latest for prompt-based data collection. Specifically, we utilized the React-based prompt to gather data, ensuring that the model could perform complex reasoning tasks. During the data collection process, whenever grounding was required to focus on specific regions of interest within images, we employed Qwen2.5VL-72B to perform the grounding tasks. This was done under the guidance of the historical trajectories.

**Data Proportions.**  To ensure that our model could perform diverse multi-step reasoning during Reinforcement Learning (RL), we carefully balanced the training data. Specifically, we balanced the trajectories based on the number of steps (2-6) and the types of actions involved (search and perception). This approach ensured that the model was exposed to a wide range of reasoning tasks and could learn to handle different types of interactions with the environment effectively.

# E  Dataset Information

We evaluate our method on three visually rich document datasets: SlideVQA, ViDoSeek, and MMLongbench.

1. **SlideVQA** [40] is a dataset for document visual question answering focused on understanding slides. It contains over 2,600 slide decks with more than 52,000 slide images and 14,500 questions that require complex reasoning skills such as single-hop, multi-hop, and numerical reasoning. The dataset is designed to support various reasoning types and includes annotated arithmetic expressions for numerical questions to enhance reasoning capabilities.

2. **ViDoSeek** [41] is a dataset specifically designed for visually rich document retrieval-reason-answer tasks. It aims to evaluate the performance of RAG systems on large-scale document collections. Unlike traditional VQA datasets that focus on single images or documents, ViDoSeek contains queries with unique answers across a collection of approximately 6,000 images, covering diverse content types such as text, charts, tables, and layouts. This dataset provides a more comprehensive and challenging benchmark for evaluating the retrieval and reasoning capabilities of RAG models in real-world scenarios.

3. **MMLongbench** [29] is a dataset designed to evaluate the document understanding capabilities of VLMs with an emphasis on long-context, multi-modal documents composed of text, images, charts, tables, and layout structures.

Table 4: Statistics of datasets.

| Dataset | Total Questions | Corpus Size | Visual Elements |
|---|---|---|---|
| SlideVQA-Test | 2020 | 8000 | Text, Chart, Table, Layout |
| SlideVQA-Train | 12268 | 44359 | Text, Chart, Table, Layout |
| ViDoSeek | 1142 | 5400 | Text, Chart, Table, Layout |
| MMLongBench | 847 | 6492 | Text, Chart, Figure, Table, Layout |

# F  Compared Baselines

Here we detailedly introduce the baselines we compare with and our re-produce details.

1. **Vanilla RAG**. There are two types of Vanilla RAG: text-based and visual-based. Text-based Vanilla RAG uses text as the retrieval corpus, which is reflected in text search engines and text modality generation. During the retrieval phase, it directly uses the original question to search for relevant text, which is then inserted into the context to answer the question. Visual-based Vanilla RAG uses images as the corpus. During the retrieval phase, it directly uses the original question to search for relevant images, which are then inserted into the context to answer the question.

2. **ReAct RAG** [48]. The method incorporates Chain-of-Thought (COT) prompting in RAG agent tasks with a format of a Thought-Action-Observation loop. The main difference between text-based and visual-based approaches lies in the retrieval corpus of the search engine and the modality of the information inserted.

3. **Search-R1** [19]. The method introduces multi-turn reasoning RL into the text RAG. We used our framework for reproducing, which includes multi-turn interactions and rule-based rewards.

4. **Search-R1-VL**. This is a vision-based baseline implemented on our framework based on search-R1. We used the same reward and post-process methods and trained models based on cold start with the same dataset as VRAG-RL.

# G  Hyperparameters

The detailed hyperparameters we use during training are shown in Table 5 and Table 6. We employ identical hyperparameters for different models.

| Table 5: Key hyperparameters for SFT. | |
|---|---|
| **Name** | **Value** |
| Finetuning type | Full |
| Freeze vision tower | True |
| Freeze multi-modal projector | True |
| Freeze language model | False |
| Cutoff len | 16384 |
| Epochs | 3 |
| Batch size | 16 |
| Gradient accumulation steps | 2 |
| Learning rate | 1.0e-5 |
| LR scheduler type | cosine |
| Warmup ratio | 0.1 |

| Table 6: Key hyperparameters for RL. | |
|---|---|
| **Name** | **Value** |
| Number of agent groups | 5 |
| Warmup steps ratio | 0.285 |
| Mini batch size | 64 |
| Micro batch size per GPU | 2 |
| Learning rate (Actor) | 1.0e-6 |
| KL loss coefficient | 0.01 (optional) |
| Tensor model parallel size | 4 |
| Total epochs | 1 |
| Max prompt length | 8192 |
| Max response length | 2048 |
| GPU memory utilization | 0.6 |

## H Case Study

In Figure 7 and 8, we list the trajectories of our VRAG-RL to illustrate how our model reasons and interacts with the environment. These cases highlight two challenges in visually rich information RAG: (1) accurately retrieving relevant images, and (2) the reference information often requires higher-resolution perception. In Figure 7, we can observe that the model demonstrated reflective capability, and eventually identified subtle clues in the relevant images. Moreover, as shown in Figure 8, the model engages in visual perception actions only when required, showcasing human-like reasoning instead of simply replicating patterns from its training data.

## I Limitations

Despite our best efforts, this paper still has some limitations: (1) Retriever Accuracy: The accuracy of the current retriever still needs improvement to better support the RAG system. (2) Base Model Capability: Enhancing the capabilities of the base model can contribute to the overall improvement of the system. (3) Real-time Requirements: The current multi-turn interactions may not meet the requirements of high real-time applications.

## J Ethics Statement

Our research complies with the NeurIPS Code of Ethics. We used publicly available datasets (as detailed in Appendix E), avoiding ethical concerns related to privacy, confidentiality, or misuse of personal biological information. The VRAG-RL framework aims to improve visual information understanding for beneficial applications. We emphasize responsible use and remain vigilant against misuse.

## K Prompts

In this section, we illustrate all the prompts used in our paper. Part of our prompts are taken from Search-R1 [19].

### K.1 Vanilla RAG Prompt

See Figure 11.

### K.2 Search-R1 Prompt

See Figure 10.

### K.3 ReAct RAG Prompt

ReAct RAG uses the same prompt as Search-R1, as shown in Figure 10.

### K.4 VRAG-RL Prompt

See Figure 9.

### K.5 Model-based Reward Prompt

See Figure 12.

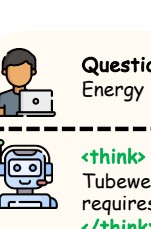

**Question:** How many more Dugwells are there than Shallow Tubewells in the years of India's Energy Divide whose map was prepared by IWMI based on Minor Irrigation Census?

**<think>** To answer this question, I need to find the specific data related to Dugwells and Shallow Tubewells in India's Energy Divide map prepared by IWMI based on Minor Irrigation Census. This requires searching for the relevant information.
**</think>**

**<search>** India's Energy Divide IWMI Minor Irrigation Census Dugwells Shallow Tubewells **</search>**

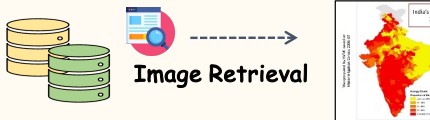

**<think>** The image provided is a map showing the proportion of electric structures in India's Energy Divide. It indicates the proportion of electric structures using different color codes. However, it does not provide specific numbers for Dugwells and Shallow Tubewells. I need to search for more detailed data on Dugwells and Shallow Tubewells from the Minor Irrigation Census.
**</think>**

**<search>** Detailed data on Dugwells and Shallow Tubewells data **</search>**

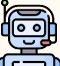

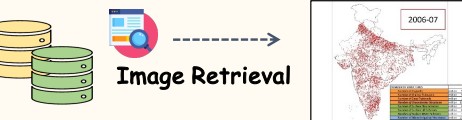

**<think>** The image contains a table with the number of structures, including Dugwells and Shallow Tubewells. I need to crop this part of the image to focus on the relevant data.
**</think>**

**<region>** [361, 366, 721, 517] **</region>**

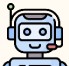

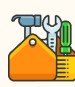

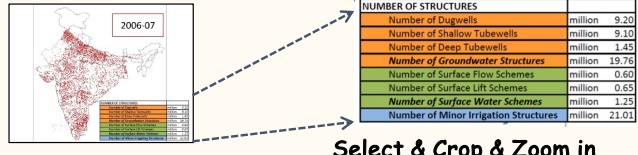

**Select & Crop & Zoom in**

**<think>** The cropped image provides the specific numbers for Dugwells (9.20 million) and Shallow Tubewells (9.10 million). To find how many more Dugwells there are than Shallow Tubewells, I need to subtract the number of Shallow Tubewells from the number of Dugwells.
**</think>**

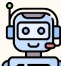

**<answer>** There are 0.10 million (or 100,000) more Dugwells than Shallow Tubewells. **</answer>**

Figure 7: Case 1 for VRAG-RL.

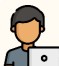

**Question:** Which H2 antagonist has a lower bioavailability: the drug that inhibits several CYP-450 isoenzymes and reduces hepatic blood flow or Famotidine?

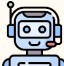

**\<think\>** To answer this question, I need to identify the H2 antagonist that inhibits several CYP-450 isoenzymes and reduces hepatic blood flow. Then, I will compare its bioavailability with that of Famotidine.
**\</think\>**
**\<search\>** H2 antagonist that inhibits several CYP-450 isoenzymes**\</search\>**

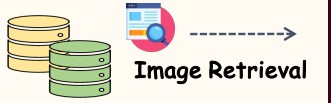

**\<think\>**The image provides information about drug interactions, specifically mentioning Cimetidine's effect on CYP-450 isoenzymes and hepatic blood flow. This aligns with the search query. I need to look at the lower right corner of the image to confirm this.
**\</think\>**
**\<region\>** [556, 337, 722, 505] **\</region\>**

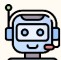

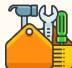

**Select & Crop & Zoom in**

**\<think\>** The diagram confirms that Cimetidine inhibits the P-450 enzymes, leading to increased serum concentrations of drugs metabolized by these enzymes. Now, I need to find information on the bioavailability of Cimetidine and Famotidine to determine which has lower bioavailability.
**\</think\>**
**\<search\>** bioavailability of Cimetidine and Famotidine **\</search\>**

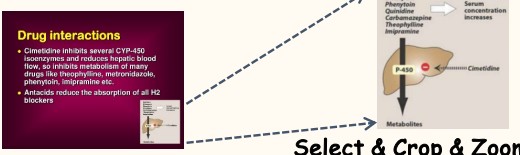

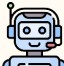

**\<think\>** Based on the provided table, Cimetidine has a bioavailability of 80%, while Famotidine has a bioavailability of 40%. Therefore, Famotidine has a lower bioavailability compared to Cimetidine. The drug that inhibits several CYP-450 isoenzymes and reduces hepatic blood flow is Cimetidine, as indicated in previous information.
**\</think\>**

**\<answer\>**Famotidine has a lower bioavailability (40%) compared to Cimetidine (80%). Therefore, the H2 antagonist with lower bioavailability is Famotidine.**\</answer\>**

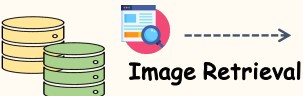

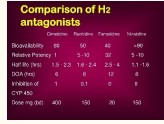

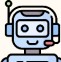

Figure 8: Case 2 for VRAG-RL.

**VRAG-RL Prompt.**

**System Prompt:**

Answer the given question. You must conduct reasoning inside <think> and </think> first every time you get new information. After reasoning, if you find you lack some knowledge, you can call a search engine by <search> query </search> and user will return the searched results. Every time you retrieve an image, you have the option to crop it to obtain a clearer view, the format for coordinates is <bbox>[x1, y1, x2, y2]</bbox>. You can search as many times as your want. If you find no further external knowledge needed, you can directly provide the answer inside <answer> and </answer>, without detailed illustrations. For example, <answer> Beijing </answer>.

- - - - - - - - - - - - - - - - - - - - - - - - - - - - - - - - - - - - - - -

**User Prompt:**
Query: {Query Description}

Figure 9: Prompt of VRAG-RL.

**Search-R1(-VL) Prompt.**

**System Prompt:**

Answer the given question. You must conduct reasoning inside <think> and </think> first every time you get new information. After reasoning, if you find you lack some knowledge, you can call a search engine by <search> query </search> and user will return the searched results. You can search as many times as your want. If you find no further external knowledge needed, you can directly provide the answer inside <answer> and </answer>, without detailed illustrations. For example, <answer> Beijing </answer>.

- - - - - - - - - - - - - - - - - - - - - - - - - - - - - - - - - - - - - - -

**User Prompt:**
Query: {Query Description}

Figure 10: Prompt of Search-R1(-VL) and ReAct RAG.

**Vanilla RAG Prompt.**

**System Prompt:**

Answer the given question. You must conduct reasoning inside <think> and </think> first every time you get new information. After reasoning, you should directly provide the answer inside <answer> and </answer>, without detailed illustrations. For example, <answer> Beijing </answer>.

- - - - - - - - - - - - - - - - - - - - - - - - - - - - - - - - - - - - - - -

**User Prompt:**
Query: {Query Description}
Reference: {Retrieved Images / Text Tokens}

Figure 11: Prompt of Vanilla RAG.

---

**Reward Model Prompt.**

**System Prompt:**

**Character Introduction**

You are an expert evaluation system for a question answering chatbot.
You are given the following information:
- the query
- a generated answer
- a reference answer

Your task is to evaluate the correctness of the generated answer.

**Response Format**

Your response should be formatted as following: <judge>True or False</judge>
If the generated answer is correct, please set "judge" to True. Otherwise, please set "judge" to False.
Please note that the generated answer may contain additional information beyond the reference answer.

- - - - - - - - - - - - - - - - - - - - - - - - - - - - - - - - - - - - -

**User Prompt:**

Query: {Query Description}
Reference Answer: {Reference Answer}
Generated Answer: {Generated Answer}

---

Figure 12: Prompt of Reward Model.

