# OpenReview forum: "VRAG-RL: Empower Vision-Perception-Based RAG for Visually Rich Information Understanding via Iterative Reasoning with Reinforcement Learning"
_NeurIPS.cc/2025/Conference — NeurIPS 2025 poster_

### Official Review · Reviewer_hXKu · 2025-06-22

**Clarity:** 2
**Significance:** 2
**Originality:** 2
**Rating:** 4
**Confidence:** 4

**Summary:**

This paper introduces VRAG-RL, a novel reinforcement learning framework for Vision-Language Models to handle visually rich information. It defines a visual perception action space for coarse-to-fine information gathering and designs a reward structure integrating retrieval efficiency, pattern outcome evaluation. Experiments on three benchmarks show VRAG-RL outperforms existing methods by 20%-30%. The framework enhances model's reasoning and retrieval capabilities through iterative interaction with search engines.

**Questions:**

1.The paper's ablation experiments on the reward design are insufficient. Table 2 does not clearly distinguish between Vanilla Reward and RAG-Specific Reward, and there are no adequate ablation experiments to demonstrate the effectiveness of the Retrieval Efficiency Reward and Model-Based Reward individually. Additionally, do the claims in 3.3 Analysis and Appendix A that the Model-Based Reward is better lack data support?

2.The paper mentioned α + β + γ = 1 in Section 2.3, but lacks specific values for α and β and balancing strategy for different reward.

**Ethical Concerns:**

["NO or VERY MINOR ethics concerns only"]

**Final Justification:**

I will keep my positive score.

**Limitations:**

The efficiency of training and inferring.

**Quality:**

2

**Strengths And Weaknesses:**

1.The paper introduces Visual Perception Actions, enabling the model to capture fine-grained visual information in images. Ablation experiments validate the effectiveness of this action, demonstrating its critical role in enhancing visual RAG capabilities.

2.The paper introduces Retrieval Efficiency Reward based on NDCG to enhance relevant image recall and reduce redundant searches.

3.Extensive experiments and ablation studies across SlideVQA, ViDoSeek and MMLongBench demonstrate performance improvements over other methods.

---

> ### Author Rebuttal · Authors · 2025-07-31
>
> Dear Reviewer hXKu,
>
> We sincerely appreciate your insightful review and the valuable insights you provided. Your constructive suggestions about the experimental analysis and reward function ablation are highly beneficial for highlighting our contributions. Your recognition of our work's strengths has been a great encouragement to our team. We have tried our best to address all concerns in the last few days. Please see our responses to each point below:
>
> > **Q1: A detailed ablation and explanation on distinguishing between Retrieval Efficiency Reward and Model-Based Reward, along with a comprehensive analysis and experimental results to support claims on Model-Based Reward.**
>
> As shown in the table below, Vanilla Reward represents the vanilla used by Search-R1, which is a rule-based outcome reward (Exact Match). Moreover, we have added more fine-grained ablation studies based on Table 2 in our manuscript to further illustrate the contribution of our retrieval efficiency reward to the overall performance improvement.
>
> | Method | Overall Performance (Acc) |
> | --- | --- |
> | Vanilla Reward | 49.3 |
> | Model-based Reward | 52.4 |
> | Model-based Reward + Retrieval Efficiency Reward | 57.1 |
>
> To further illustrate the impact of our RAG-specific reward on overall retrieval performance, we present more detailed experimental results in the table below. The retrieval performance gap between w/ Retrieval Efficiency Reward (α=0.2) and w/o Retrieval Efficiency Reward (α=0) serves as a supplement to Figure 4 in out manuscripts, further demonstrating the effectiveness of our module.
>
> | Method | Retrieval Performance (Recall@1) |
> | --- | --- |
> | w/o Retrieval Efficiency Reward | 55.9 |
> | w/ Retrieval Efficiency Reward | 60.3 |
>
> In order to support claims on Model-Based Reward, we decompose the model evaluation into three aspects: Invalid Action Rate, Finish Rate, and End-to-End Performance, as shown in the table below.
>
> | Method | Invalid Action Rate $\downarrow$ | Finish Rate $\uparrow$ | Overall Performance $\uparrow$ |
> | --- | --- | --- | --- |
> | RL w/ Rule-based Reward | 9.2 | 81.8 | 49.3 |
> | RL w/ Model-based Reward | 6.7 | 94.2 | 52.4 |
>
> In our agentic RAG task, the training effect of rule-based reward affects the model's performance by influencing the sampling distribution and sampling efficiency. Specifically, rule-based rewards make the model prefer learning shorter answers and shorter trajectories (which can easily achieve higher rewards) because rule-based rewards cannot effectively evaluate longer and more complex answers. However, in the test set and real-world environment, the model's final Finish Rate may decrease, and the Invalid Action Rate may increase.
>
> Apart from the experimental results, during training we analyzed the advantages of model-based rewards compared to rule-based rewards (Recall/Exact Match/ANLS), which are mainly reflected in: (i) The practicality of the final model: The model provides answers in the form of complete sentences or paragraphs rather than short answers consisting of one or two words; (ii) Dataset adaptability: The current datasets contain multiple key points instead of phrases or words, and using model-based rewards allows training on more datasets; (iii) Rule-based rewards are difficult to evaluate at the semantic level; (iv) The model is more prone to hacking rule-based rewards, which can ultimately lead to repetitive outputs or training collapse.
>
> > **Q2: About the specific values for α and β, and the strategy for balancing different rewards.**
>
> We experimented with many combinations of weights and conducted quantitative analysis during our research.
>
> $r_\phi = \alpha \cdot r_{Ret} + \beta \cdot r_{Ans} + \gamma \cdot r_{Pat}$
>
> The experimental results are shown in the table below.
>
> | α | β | Overall Performance |
> | --- | --- | --- |
> | 0.1 | 0.8 | 53.9 |
> | 0.2 | 0.7 | 57.1 |
> | 0.3 | 0.6 | 55.4 |
>
> In practice, for the parameter γ, it can be set to a small weight like 0.1 when the model's instruction-following ability is good after SFT. During the actual implementation, when extracting the final answer and retrieval results, we will conduct a pattern validity check.
>
> For the parameters α and β, it is better to prioritize the final answer reward and adjust the parameter values within a smaller range. This conclusion can be drawn from observing the results of the first and last lines. When α is too large, performance decreases, as the model tends to search multiple times, resulting in lower confidence in the answers provided. Conversely, when α is too small, the insufficient activation of the model's search capabilities can lead to suboptimal results. We chose the parameter setting of α=0.2 and β=0.7 for the experiment.

---

> > ### Comment · Reviewer_hXKu · 2025-08-05
> > **res**
> >
> > I thank the authors for their response and I maintain a positive view.

---

> > > ### Author Response · Authors · 2025-08-05
> > > **Thanks for the Positive Feedback and Recognition of Our Work**
> > >
> > > Dear Reviewer hXKu,
> > >
> > > We sincerely appreciate your insightful review and the valuable insights you provided. We have incorporated these insights into our revision plan and believe the updated manuscript will better serve the research community.
> > >
> > > Your recognition of our work's strengths has been a great encouragement to our team!
> > >
> > > We deeply value your time and expertise in reviewing our work. **If possible, we would kindly like to ask if there is any opportunity to raise the overall score in this paper in light of these updates. Thank you again for your valuable insights and for your consideration!**
> > >
> > > Warm regards,
> > >
> > > The Authors

---

### Official Review · Reviewer_8qoA · 2025-06-30

**Clarity:** 3
**Significance:** 3
**Originality:** 3
**Rating:** 5
**Confidence:** 4

**Summary:**

This paper proposes VRAG-RL, a reinforcement learning approach that trains Vision Language Models to either (1) generate an answer; (2) produce a visual perception action (i.e., cropping); and (3) produce a query to be sent to a search engine. These actions are preceded by instructed "thinking" as in a ReAct manner.

The authors conduct experiments on ViDoSeek, SlideVQA, and MMLongBench with QWen2.5-VL-3B/7B-Instruct models. They show that the VRAG-RL approach performs better than vanilla RAG, ReAct, and Search-R1 substantially.

**Questions:**

* It seems like only one image is retrieved/cropped in one particular turn, is it possible to extend to multiple images?
* How is the performance of vanilla RAG/ReACT/Search-R1-VL when multiple images are provided at each turn?
* It seems like there is no explicit supervision signal for the visual perception action (cropping) except for the formatting reward. What experimental results suggest that the model is correctly identifying the region of interests other than end-to-end performance? Could you make a case that VRAG-RL is learning to effectively identify region-of-interest?
* In Figure 6, what’s the x-axis ticks?
* In Figure 5, what's the scale of the radar chart?

**Ethical Concerns:**

["NO or VERY MINOR ethics concerns only"]

**Final Justification:**

I maintain a positive view on the strength of the paper and the author's response.

**Limitations:**

yes

**Paper Formatting Concerns:**

None.

**Quality:**

3

**Strengths And Weaknesses:**

Strengths:
* The inclusion of visual perception actions (i.e., cropping region-of-interest), is novel and suitable for the task.
* The authors show RL can effectively train models to produce visual perception actions along with search queries, which provides a strong baseline for the field.
* The paper is clearly written.

Weaknesses:
* Clarity of some results. Figure 6 is missing the x-axis ticks. Figure 5 is not interpretable as there are no scales to the radar chart.
* Discussion of Top-K retrieval. In RAG approaches it is common to retrieve topK documents to answer the question. It seems like the paper focuses on the Top-1 case in each turn. It is not clear how the approach could extend to the more general TopK scenario. It is also unclear if baseline approaches are all implemented in the Top-1 setting. If so, the comparison could be improved as it is straight forward to extend to the Top-K setting in Vanilla RAG/ReAct (the QWen2.5-VL model family supports that). This could be a limitation for the VRAG-RL approach, but is not discussed by the author.

---

> ### Author Rebuttal · Authors · 2025-07-31
>
> Dear Reviewer 8qoA,
>
> We sincerely appreciate your insightful reviews and constructive suggestions, which are highly beneficial for  enhancing the comprehensiveness of our experimental results and the readability of our manuscripts. Your recognition of the strengths of our work has been a great encouragement to our team. We have made every effort to address all concerns in the past few days. Please see our responses to each point below:
>
> > **W1&Q4: Clarification regarding Fig. 6.**
>
> For Figure 6, the x-axis represents the average action count, with scales labeled as 1, 2, 3, ... We use the number of answer actions to normalize the count of other actions.
>
> From a macro perspective, introducing visual perception actions can make reasoning more efficient, thereby avoiding excessively long and repetitive retrievals. We will update the revision to make the chart more readable according to your valuable suggestions.
>
> > **W1&Q5: Clarification regarding Fig. 5.**
>
> For Figure 5, the scales of the radar chart represent the normalization of performance relative to the state-of-the-art method (with Visual Perception). Each concentric circle increases by 25% from the center outward.
>
> This chart indicates that visual perception action indeed provides the model with a fine-grained perspective for visually rich information understanding. We will update the revision to enhance the clarity of the figures according to your valuable suggestions.
>
> > **W2&Q1: How VRAG-RL can be extended to handle top-k inputs as multiple images.**
>
> From a macro perspective, for the entire trajectory, VRAG-RL can be seen as sequentially inputting multiple images, making judgments and scaling each image individually. In contrast, inputting multiple images at once can be considered batch inference. Our current sequential process paradigm, which involves reviewing images one by one, can be regarded as a paradigm of test-time scaling.
>
> We greatly appreciate your insightful perspectives for the multiple images in each turn. We have attempted to extend our Visual Perception Action to support multiple image inputs in a simple yet effective manner: Stitch multiple images together and then let the model output the region of interest. When images are stitched together and input as a single image (with the model being informed that the images are arranged from top to bottom, left to right), the model can identify each image and output the regions of interest at once.
>
> In the experiment, we attempted input formats including 1x2, 1x3, and 2x2. We employed our original sequential paradigms to prompt the model to select the region of interest of the stitched images. The experimental results are shown in the table below.
>
> | Paradigms | Comment | Top2 (1x2) | Top3 (1x3) | Top4 (2x2) |
> | --- | --- | --- | --- | --- |
> | Select Region of Interest -> Answer | Align with training paradigm | 58.4 | 54.1 | 60.1 |
>
> By analyzing the case, we think that the 1x3 method causes the input image ratio to differ from the pre-training domain, resulting in a decrease in the model's perception capability. In practical use, we recommend image ratios that are more balanced in terms of length and width, such as 1x2 or 2x2.
>
> > **W2&Q2: The performance of vanilla RAG/ReACT/Search-R1-VL when multiple images are provided at each turn.**
>
> Below are the experimental results for multiple images provided at each iteration. All experiments utilize the Qwen2.5VL model with the same retrieval corpus, as shown in the table below.
>
> | Method | Top2 | Top3 | Top4 |
> | --- | --- | --- | --- |
> | Vanilla RAG | 32.5 | 36.1 | 37.2 |
> | ReAct | 33.4 | 35.2 | 36.9 |
> | Search-R1-VL | 40.2 | 46.1 | 49.4 |
>
>
> For the baseline method, we conducted experiments by directly changing the number of input images in each turn. Regarding prompt engineering based methods, when comparing Vanilla RAG and ReAct RAG, we observed that as the number of input images increased, Vanilla demonstrated stronger performance than ReAct. This is because the excessively long context hinders the model's ability to follow instructions, making it challenging to provide coherent tool calls or effective final answers. However, Search-R1-VL addresses this issue; the trained model has a better instruction-following capability. Nonetheless, due to the low density of visual information, there is still a performance gap compared to VRAG-RL.
>
>
>
> > **Q3: A more in-depth discussion about VRAG-RL is learning to effectively identify region-of-interest.**
>
> We can compare the consistency between the trajectory annotations made by mixed experts and the regions of interest output by VRAG-RL on the same dataset to verify whether the model is learning to identify the regions of interest. The experimental steps are as follows:
>
> (i) Use mixed-expert sampling and VRAG-RL to perform inference and obtain trajectories respectively.
>
> (ii) For each identical query, we match the same images retrieved and regions of interest within the trajectories.
>
> (iii) Calculate the mAP of the region of interest in the matched images.
>
> We evaluate the performance of the models after SFT and RL separately. The experimental results are shown in the table below.
>
> | Method | mAP@50 $\uparrow$ |
> | --- | --- |
> | SFT | 43.1 |
> | +RL | 49.2 |
>
> We used metrics from object detection to verify whether the model can effectively learn to recognize regions of interest. In the experiment mentioned above, we validated that our model gradually aligns with the output distribution of large-scale expert models. The model's ability to identify regions of interest progressively improved, confirming the effectiveness of our approach in reducing model hallucinations.
>
> Although there is no supervision for region coordinates in our reward, actually, if the model can correctly focus on the regions of interest, the final answer will also be correct. The reward for the final answer is consistent with the model's ability to identify the region of interest, forming a supervisory signal for the model.

---

> > ### Comment · Reviewer_8qoA · 2025-08-04
> >
> > I thank the authors for their response. Adding these new results to the paper in revision can strengthen the paper (the TopK experiments in particular). I maintain a positive view.

---

> > > ### Author Response · Authors · 2025-08-05
> > > **Thanks for the Positive Feedback and Recognition of Our Work**
> > >
> > > Dear Reviewer 8qoA:
> > >
> > > We appreciate your time and effort in reviewing our manuscript. We are very grateful for the valuable comments and suggestions you have provided. They have been of great help to us in improving our manuscript.
> > >
> > > Thank you for your patience and recognition.
> > >
> > > Best regards and thanks,
> > >
> > > Submission9287 Authors.

---

### Official Review · Reviewer_jJNP · 2025-07-02

**Clarity:** 3
**Significance:** 2
**Originality:** 2
**Rating:** 5
**Confidence:** 3

**Summary:**

The paper proposes VRAG-RL, a reinforcement learning framework to enhance vision-language models (VLMs) for retrieval-augmented generation (RAG) tasks involving visually rich documents. It introduces a visual perception action space and a fine-grained reward function combining retrieval efficiency and answer quality.

**Questions:**

- The paper employs Discounted Cumulative Gain (DCG) as part of the retrieval reward. While the results show its importance, the motivation for choosing DCG is not thoroughly discussed. Why DCG? Does it empirically outperform alternatives, or was it selected for simplicity?
- The authors use a re-annotated dataset for VRAG-RL’s SFT/RL training, while baselines (Vanilla RAG, ReAct, Search-R1) likely use original annotations. Does the performance gap in Table 1 partly reflect data quality/quantity differences rather than method superiority?
- The paper emphasizes RL’s role in optimizing retrieval and reasoning, but how much does VRAG-RL improve over an SFT-only version?
- The reward coefficients (α, β, γ in Eq. 12) are set heuristically. Was their sensitivity tested?

**Ethical Concerns:**

["NO or VERY MINOR ethics concerns only"]

**Final Justification:**

After carefully reviewing the authors' response and the discussion among reviewers, I acknowledge that the technical components of this work are generally solid. However, my concern about the paper's core contribution remains unresolved. While the individual modules are well-motivated and technically sound, they appear to address somewhat separate challenges, and the paper could benefit from a more unified perspective to better highlight its overarching insight. This currently makes it challenging to pinpoint the paper's key message.
As a result, I have decided to increase my score and lower my confidence.

**Limitations:**

Yes

**Paper Formatting Concerns:**

No formatting issue

**Quality:**

3

**Strengths And Weaknesses:**

### Strength
- The proposed VRAG-RL achieves significant improvements over baseline methods
- The visual perception action space (cropping/zooming regions) allows VLMs to focus on information-dense areas

### Weakness
- Some key components, such as cropping/resizing as visual perception actions and the think-then-answer (ReAct-style) paradigm, are not fundamentally novel. Similar ideas have appeared in prior work (e.g., Search-R1). While the integration of these techniques is well-executed, the paper does not clearly articulate what makes its core formulation (e.g., action space design) uniquely innovative compared to existing approaches.
- The paper introduces multiple modules (visual perception actions, retrieval reward, re-annotated dataset, multi-round generation), but it’s unclear which one is the primary contribution.

---

> ### Author Rebuttal · Authors · 2025-07-31
>
> Dear Reviewer jJNP,
>
> We sincerely appreciate your insightful review and the valuable insights you provided. Your recognition of our work's strengths has been a great encouragement to our team. We have tried our best to address all concerns in the last few days. Please see our responses to each point below:
>
> > **W1: A more detailed explanation of what makes the core formulation (e.g., action space design) uniquely innovative compared to existing approaches (e.g., Search-R1) would be beneficial.**
>
> Although recent works like Search-R1 have utilized ReAct-style in textual tasks, when generalizing to the multimodal tasks, visual-related actions are required. Teaching VLMs these actions is fundamentally different from LLM-based methods. On the other hand, while some current works include crop/resize functions, generalizing them to multi-round interactions and multiple image inputs requires an entirely new paradigm and redesigned data construction pipeline. Below, we will highlight our core innovations one by one.
>
> + VRAG-RL v.s ReAct-Style Agentic RAG (Search-R1)
>     - Differences: Search-R1 transitions RL from single-turn to multiple interactions, while our work comprehensively migrates RL to multimodal, multi-turn agentic tasks and provides insightful optimizations for RAG tasks.
>     - Our core innovation: (i) Visual-specific action space. In a multi-round scenario, we propose multi-image visual action, which is a completely new paradigm. (ii) RAG-specific reward and trajectory-wise evaluation: We decompose the model's capabilities into retrieval and reasoning. Our reward is calculated based on the entire trajectory, including both retrieval efficiency and the quality of the final answer.
> + VRAG-RL v.s VLMs with Crop/Grounding/Resize
>     - Differences: The current work is just a single input without interaction, and it only allows for the one image QA task. Our work introduces a novel paradigm called visual specific retrieval-augmented reasoning, which supports multi-turn interaction and multi-image input.
>     - Our core innovation: (i) Scalable multi-expert data construction: We propose Multi-Expert Sampling, which efficiently constructs high-quality multi-turn trajectories by leveraging different capabilities of the expert model. (ii) Multi-modal Multi-turns Training Framework: Recent methods focus only on the understanding of single images, while we propose a novel multimodal multi-round RL training framework from scratch.
>
> We will update the above discussion in revision according to your insightful review to highlight our contribution, providing more potential insights for the whole community.
>
> > **W2: About the primary contribution in multiple modules proposed.**
>
> Our core idea is to optimize the multimodal agentic RAG task. As the capabilities of the RAG system can be decomposed into retrieval and reasoning, we will detail our key contribution to enhancing these two capabilities by discussing problem identification, the proposed approach, and implementation, as illustrated in the table below.
>
> | Core Contributions | Problem Identification | Novel paradigm/mechanism | Implementation Approach |
> | --- | --- | --- | --- |
> | Multi-modal agentic RL framework | Lack of an effective multimodal, multi-turn RL training framework. | Multimodal multi-round sampling and optimization strategy. | Implemented using GRPO which optimizes the model’s retrieval and reasoning capability. |
> | Novel paradigm for iterative visual reasoning | The text-based reasoning overlooks the sensitivity of visual resolution and information density. | Visual perception action space, allowing VLMs to gather information from coarse-grained to fine-grained. | Multi-Expert data construction pipeline to teach the model the new paradigm. |
> | Fine-grained RAG-specific training strategy | Current agentic training rely solely on outcome rewards. | Comprehensive reward based on the entire trajectory evaluation. | Reward that integrates retrieval efficiency and outcome reward. |
>
>
> In summary, the multiple modules we propose are closely centered around three insights. Each core contribution addresses a key issue through one or more modules.
>
> > **Q1: Why DCG? Does it empirically outperform alternatives, or was it selected for simplicity?**
>
> We have explored different metrics based on retrieval performance for reward. As described in Section 2.3, our pilot study indicates that retrieving relevant documents earlier and more completely is beneficial for correctly answering questions. When relevant information cannot be retrieved, the model is unable to produce the correct answer. Even when the model finds relevant information through multiple rounds of search, overly lengthy context can still lead to suboptimal results. Therefore, we need a metric that can comprehensively evaluate retrieval efficiency and completeness to help optimize the model. The table below presents our analysis of various metrics:
>
> | Metric | Focus | Usage scenarios of metrics | Comment |
> | --- | --- | --- | --- |
> | Recall  | Coverage of relevant documents | When minimizing missed relevant documents | Unsuitable. Only include coverage. |
> | MRR | Efficiency of finding the  relevant document	 | When quickly finding the relevant information matters | Unsuitable. Only focus on the ranking without comprehensiveness. |
> | NDCG | Ranking quality with relevance scores	 | When considering both relevance and ranking quality | Suitable. Considering both retrieval efficiency and accuracy. |
>
>
> Recall is based solely on the retrieval results and do not include evaluation of ranking and efficiency. MRR (Mean Reciprocal Rank) only considers the ranking of documents without taking into account those that have not been retrieved. However, we need to consider retrieving every relevant document.
>
> NDCG takes into account both the ranking and accuracy of all relevant documents, making it suitable as a reward for evaluating the efficiency and comprehensiveness of the agentic RAG. In addition, we conducted experiments on VRAG-7B, and the results are shown as follows.
>
> | Method | Overall Performance |
> | --- | --- |
> | Model-based Reward | 52.4 |
> | w/ Recall-based Reward | 53.7 |
> | w/ MRR-based Reward | 53.2 |
> | w/ NDCG-based Reward (Ours) | 57.1 |
>
>
> As shown in the results, the reward based on Recall and MRR offers a slight gain. However, for RAG-specific tasks, our NDCG-based reward is more suitable as it provides a comprehensive evaluation of retrieval performance.
>
> > **Q2: Regarding the performance gap in the main results shown in Table 1, do they reflect the advantage of the data or the advantage of the method?**
>
> For the performance of large models, both data and methods are equally important. Our work not only proposes methods for constructing high-quality data but also introduces novel and effective paradigms.
>
> Below, we will demonstrate the effectiveness of our method with additional experiments:
>
> (i) Demonstrate the effectiveness of our method using the same training data: VRAG-RL and Search-R1-VL are trained with the same data of QA pairs. The performance gap shown in Table 1 of our manuscripts demonstrating the effectiveness of our framework.
>
> (ii) Demonstrate the effectiveness of our proposed paradigm by prompting the same model: We separately evaluate the trajectories constructed using the ReAct-based pipeline and the trajectories constructed using our visual perception paradigm, both with the same model. Through this experiment, we can evaluate the effectiveness of our paradigm without the interference of training data. The experimental results are shown in the table below.
>
> | Method | Accuracy (%) |
> | --- | --- |
> | Vanilla Pipeline | 58.9 |
> | Visual Perception Pipeline | 67.2 |
>
> The experiments and analysis above have demonstrated the effectiveness of our method through controlling variables. We will update in the revision according to your highly valuable suggestions.
>
> > **Q3: How much does VRAG-RL improve over an SFT-only version before RL?**
>
> The multimodal RL paradigm we proposed can achieve significant gains, as shown in the experimental results in the table below.
>
> | Method | Invalid Action Rate $\downarrow$ | Finish Rate $\uparrow$ | Overall Performance $\uparrow$ |
> | --- | --- | --- | --- |
> | SFT | 9.4 | 84.2 | 48.9 |
> | +RL | 5.1 | 97.1 | 57.1 |
>
>
> According to the results, we believe that the improvement in end-to-end performance mainly stems from the understanding and utilization of tools. In fact, how agentic RAG retrieves more relevant information and avoids repetitive retrievals is the primary issue that needs to be addressed. By effectively utilizing tools, the model can then perform effective reasoning, which is also the fundamental factor that defines its performance upperbound. This is also explained in Section 3.3 and Table 3 of our manuscripts.
>
> > **Q4: About the sensitivity of performance to the α/β/γ weights.**
>
> We experimented with many combinations of weights and conducted quantitative analysis during our research.
>
> $r_\phi = \alpha \cdot r_{Ret} + \beta \cdot r_{Ans} + \gamma \cdot r_{Pat}$
>
> The experimental results are shown in the table below.
>
> | α | β | γ | Overall Performance |
> | --- | --- | --- | --- |
> | 0.1 | 0.8 | 0.1 | 53.9 |
> | 0.2 | 0.7 | 0.1 | 57.1 |
> | 0.3 | 0.6 | 0.1 | 55.4 |
>
>
> In practice, for the parameter γ, it can be set to a small weight like 0.1 when the model's instruction-following ability is good after SFT. During the actual implementation, when extracting the final answer and retrieval results, we will conduct a pattern validity check.
>
> For the parameters α and β, when α is too large, performance decreases, as the model tends to search multiple times, resulting in lower confidence in the answers provided. Conversely, when α is too small, the insufficient activation of the model's search capabilities can lead to suboptimal results. We chose the parameter setting of α=0.2 and β=0.7 for the experiment.

---

> > ### Comment · Reviewer_jJNP · 2025-08-05
> >
> > I appreciate the authors' response and remain positive about the work

---

> > > ### Author Response · Authors · 2025-08-06
> > > **Thanks for the Positive Feedback and Recognition of Our Work**
> > >
> > > Dear Reviewer jJNP:
> > >
> > > Thank you so much for the recognition of our responses. We are glad to your positive feedback! Thanks!
> > >
> > > Following your constructive suggestions, we will make more efforts to improve our paper further. The modifications and analysis in our response are summarized as follows:
> > >
> > > - We conducted a detailed discussion and outlined the key innovations of our approach to highlight our contributions.
> > > - We conducted a more detailed data analysis to demonstrate the effectiveness of our proposed training methods and visual thinking paradigms.
> > > - We compared various retrieval metrics to explain our choice of NDCG and conducted additional experiments to demonstrate the effectiveness of our reward, aiming to provide more potential insights for the community.
> > >
> > > We deeply value your time and expertise in reviewing our work. **If possible, we would kindly like to ask if there is any opportunity to raise the overall score in this paper in light of these updates. Thank you again for your valuable insights and for your consideration!**
> > >
> > > Many thanks for your constructive comments, time, and patience.
> > >
> > > Best regards and thanks,
> > >
> > > The Authors

---

### Official Review · Reviewer_FHAe · 2025-07-03

**Clarity:** 3
**Significance:** 3
**Originality:** 2
**Rating:** 5
**Confidence:** 3

**Summary:**

The paper proposes VRAG-RL, a reinforcement learning framework designed to enhance vision–language models (VLMs) by introducing a visual-perception action space (e.g., cropping and zooming) and a composite reward tailored for retrieval-augmented generation (RAG). This reward combines retrieval efficiency, answer correctness based on a model, and consistency with common patterns. The training approach combines supervised fine-tuning from multiple experts with reinforcement learning using a GRPO-based policy gradient. Evaluated on three visually complex QA benchmarks (SlideVQA, ViDoSeek, and MMLongBench) VRAG-RL improves top-line accuracy for a 3B model and a 7B model, compared to the strongest baseline, Search-R1-VL.

**Questions:**

- Have you tried reward-model swap experiments (e.g., using GPT-4o judge) to verify that gains persist?
- How sensitive is performance to the α/β/γ weights in Eq 12?
- Could the visual perception policy hallucinate crops (i.e., propose empty regions)? What proportion of invalid crops remain after RL?

**Ethical Concerns:**

["NO or VERY MINOR ethics concerns only"]

**Final Justification:**

Thank you for the strong rebuttal. The additional experiments, failure case analysis, and Vision-R1 comparison effectively address my concerns and enhance the work’s clarity and rigor. I am therefore increasing my rating.

**Limitations:**

Yes

**Quality:**

3

**Strengths And Weaknesses:**

Strengths:
- Well-motivated visual action space that reduces context and raises “effective resolution” without inflating input length
- Clear empirical gains on three diverse benchmarks and two model sizes. Ablations show that each component matters.
- End-to-end agent framework (rollout algo 1) that unifies search, cropping, and answer steps; cases show coherent multi-turn reasoning.

Weaknesses:
- Because both training and evaluation depend on the same 7B Qwen reward model (binary accuracy), the policy may learn to exploit that grader’s characteristics rather than the underlying task. Adding independent checks (like exact numeric accuracy on SlideVQA) might reduce reward hacking and improve the results.
- Paper highlights two success cases but does not discuss the failure patterns (e.g., when cropping the wrong region, retrieval recall is low, etc.). Understanding failure modes would guide future work.
- Recent vision-RL agents such as Vision-R1 and Perception-R1 (2025) are not included. Search-R1-VL is a fair baseline, but comparison to those highly-related methods is essential to judge novelty and delta.

---

> ### Author Rebuttal · Authors · 2025-07-31
>
> Dear Reviewer FHAe,
>
> We sincerely appreciate your insightful review and the valuable insights you provided, which are highly beneficial for highlighting our contributions. Your recognition of our work's strengths has been a great encouragement to our team. We have tried our best to address all concerns in the last few days. Please see our responses to each point below:
>
> > **W1: Adding independent checks (like exact numeric accuracy on SlideVQA) might reduce reward hacking and improve the results.**
>
> Following your constructive suggestions, we perform consistency checks between our current reward strategy and the rule-based evaluation strategy to verify that reward hacking does not exist. As these two evaluation strategies are completely different, when both strategies show the same performance improvement trend, it indicates that our model is objectively being optimized, rather than relying on hacking the reward. The results are shown in the table below.
>
> | Method | F1 | Accuracy (Binary) |
> | --- | --- | --- |
> | Vanilla RAG | 29.4 | 25.9 |
> | ReAct | 37.1 | 30.9 |
> | Search-R1-VL | 42.9 | 46.7 |
> | VRAG-RL | 53.2 | 62.3 |
>
>
> We used the F1 metric from the SlideVQA benchmark for additional evaluation. The Pearson coefficient between the two sets of data is calculated to be 0.97, which indicates a strong positive linear correlation between the two metrics. The consistency in their trends demonstrates the effectiveness of our reward and the improvement of our framework.
>
>
>
> > **W2: Analyzing more failure cases (e.g., when cropping the wrong region, retrieval recall is low, etc.) would guide future work.**
>
> Thank you for your highly valuable suggestions, which are beneficial in highlighting our contributions. Besides simply giving incorrect answers, we categorize failure patterns into three types and we also provide corresponding solutions to each type of failure case. Some of these issues are commonly encountered by agentic RAG, while others are unique to our VRAG-RL, as follows:
>
> + The coordinates of the region of interest have a slight bias.
>     - Analysis & Solution: The grounding task in the pre-training domain of Qwen2.5-VL uses actual pixel format (some VLMs use normalized coordinates). Since the patch size in vision embedding is 28*28, a slight bias is quite normal. During training, we use the original coordinates. But during inference, we perform region upsampling as $[\lfloor x_{min} / 28 \rfloor \cdot 28, \lfloor y_{min} / 28 \rfloor \cdot 28,\lceil x_{max} / 28 \rceil \cdot 28,\lceil y_{max} / 28 \rceil \cdot 28]$. This simple yet effective approach slightly expands the region to an integer multiple of the patch size, effectively mitigating the localization hallucination without any additional overhead.
> + The model selection for cropping regions is irrelevant (or empty).
>     - Analysis & Solution: During the SFT, we scale up more high-quality data by mixing experts and  increase the proportion of visual perception action data. During the RL, we enhance this capability of action. Generally speaking, the trajectory that answers correctly is highly related to the golden reference in its region of interest, which is one of the reasons why reinforcement learning is effective.
> + Unable to retrieve relevant information or too many search iterations, which is a common issue with agentic RAG.
>     - Analysis & Solution: In our approach, we utilize the Retrieval Efficiency Reward to optimize the model's interaction with search engines, addressing this issue to a certain extent. In practice, due to the limitations of search engine performance, it often fails to retrieve relevant information. When the interaction process becomes too lengthy, we prompt the model to provide answers by summarizing existing information, such as "Please give me the answer based on context in \<answer\>...<\answer>." This method enables the model to offer a possible answer even when its confidence is not high.
>
> We will discuss these failure patterns and solutions along with more specific cases in the revision, to highlight our contributions and provide potential inspiration for future work.
>
>
>
> > **W3: Comparing with more recent works such as vision-r1 and perception-r1 would help highlight the novelty.**
>
> Following your highly valuable suggestions, we have summarized the latest relevant work, as shown in the table below.
>
> | Method | Task | Context | Action Space | Multi-Turn Interaction |
> | --- | --- | --- | --- | --- |
> | Perception-R1 (2025.4) | Detection&OCR  | Single Image | None | ✗ |
> | VLM-R1 (2025.4) | Detection | Single Image | None | ✗ |
> | Vision-R1 (2025.4) | Mainly focus on mathematical geometry QA | Single Image | None | ✗ |
> | VRAG-RL (Ours) | open-domain QA | Multi Images | Visual Perception | ✔ |
>
>
> During the rebuttal, we reviewed the latest visual perception related works from preprints over the past three months. Perception-R1 and VLM-R1 are both trained on the Grounding task, with outputs constrained to coordinates and object categories/contents, and evaluation metrics using mIoU, which cannot be transferred to evaluate our general QA task. Vision-R1 is trained on mathematical geometry task and can generalize to general QA tasks. Therefore, within related work above, we can only conduct more comparative experiments with Vision-R1, as shown in the table below.
>
> | Method | Accuracy |
> | --- | --- |
> | Vision-R1 | 24.4 |
> | VRAG-RL | 57.1 |
>
>
> The Vision-R1 model performs well in single image QA tasks, but its performance is constrained by its limited ability to handle multi-turn tool calls and multi-image understanding. In fact, when it comes to agentic RAG tasks, the model's capabilities expand beyond mere image understanding to include effective retrieval and tool call features. Considering the characteristics of the agentic RAG task, it is essential to optimize both the retrieval and understanding capabilities of the model, which also highlights our core contribution.
>
> > **Q1: Have you tried reward-model swap experiments (e.g., using GPT-4o judge) to verify that gains persist?**
>
> During the rebuttal, we conducted additional experiments using GPT-4o as a reward to verify the gains. In our experiments, the Qwen2.5 model with 7B parameters was already capable of determining semantic correctness based on the response and golden answer. The api-based large scale model such as GPT-4o has achieved human-level performance in this simple task. The experiments are shown in the table below.
>
> | RL Reward Model | Overall Evaluation Results |
> | --- | --- |
> | GPT-4o | 57.3 |
> | Qwen2.5-7B-Instruct | 57.1 |
>
>
> From the experimental results, the gain is still consistent across different experimental settings, which validates the effectiveness of our original setting. Indeed, as long as the model is provided with sufficiently clear prompts, it becomes quite simple to judge whether an answer is correct or incorrect. This is also the reason why Qwen-7B can effectively serve as our reward model.
>
> > **Q2: How sensitive is performance to the α/β/γ weights in Eq 12?**
>
> We experimented with many combinations of weights and conducted quantitative analysis during our research.
>
> $r_\phi = \alpha \cdot r_{Ret} + \beta \cdot r_{Ans} + \gamma \cdot r_{Pat}$
>
> The experimental results are shown in the table below.
>
> | α | β | γ | Overall Performance |
> | --- | --- | --- | --- |
> | 0.1 | 0.8 | 0.1 | 53.9 |
> | 0.2 | 0.7 | 0.1 | 57.1 |
> | 0.3 | 0.6 | 0.1 | 55.4 |
>
>
> In practice, for the parameter γ, it can be set to a small weight like 0.1 when the model's instruction-following ability is good after SFT. During the actual implementation, when extracting the final answer and retrieval results, we will conduct a pattern validity check.
>
> For the parameters α and β, it is better to prioritize the final answer reward and adjust the parameter values within a smaller range. This conclusion can be drawn from observing the results of the first and last lines. When α is too large, performance decreases, as the model tends to search multiple times, resulting in lower confidence in the answers provided. Conversely, when α is too small, the insufficient activation of the model's search capabilities can lead to suboptimal results. Ultimately, we chose the parameter setting of α=0.2 and β=0.7 for the experiment.
>
> > **Q3: About the hallucination of visual perception policy and invalid action rate after RL.**
>
> Our training method effectively teaches the model to identify regions of interest and significantly enhances the model's ability to perform effective actions, as shown in the table below, which is also explained in Table 3 of our manuscript.
>
> | Method | Invalid Action Rate $\downarrow$ | Finish Rate $\uparrow$ |
> | --- | --- | --- |
> | SFT | 9.4 | 84.2 |
> | +RL | 5.1 | 97.1 |
>
>
> We can also compare the consistency between the trajectory annotations made by mixed experts and the regions of interest output by VRAG-RL on the same dataset to verify whether the model is learning to identify the regions of interest. The experimental steps are as follows:
>
> (i) Use mixed-expert sampling and VRAG-RL to perform inference and obtain trajectories respectively.
>
> (ii) For each identical query, we match the same images retrieved and regions of interest within the trajectories.
>
> (iii) Calculate the mAP of the region of interest in the matched images.
>
> We evaluate the performance of the models after SFT and RL separately. The experimental results are shown in the table below.
>
> | Method | mAP@50 $\uparrow$ |
> | --- | --- |
> | SFT | 43.1 |
> | +RL | 49.2 |
>
> In the experiment mentioned above, we validated that our model gradually aligns with the output distribution of large-scale expert models. The model's ability to identify regions of interest progressively improved, confirming the effectiveness of our approach in reducing model hallucinations.

---

> ### Author Response · Authors · 2025-08-06
> **Welcome for Further Feedback!**
>
> Dear Reviewer FHAe:
>
> Thank you so much for the recognition of our responses. We are glad to your positive feedback! Thanks!
>
> Following your constructive suggestions, we will make more efforts to improve our paper further. During the rebuttal, we have performed additional experiments and more analysis to address your concerns. The modifications and analysis in our response are summarized as follows:
>
> - We discussed the latest work from the past three months (Perception-R1 \ VLM-R1 \ Vision-R1) and conducted additional experiments to demonstrate the effectiveness of our VRAG-RL in the RAG task.
> - We conducted a more detailed data analysis focusing on ablation, model behavior, and sensitivity to hyperparameters.
> - We discussed common failure cases and analyzed the reasons from aspects such as model architecture and training data construction. We provided our solutions and ideas to offer more potential insights to the community.
>
> We deeply value your time and expertise in reviewing our work and wonder if the responses have addressed your concerns. **If possible, we would kindly like to ask if there is any opportunity to raise the overall score in this paper in light of these updates. Thank you again for your valuable insights and for your consideration!**
>
> Many thanks for your constructive comments, time, and patience.
>
> Best regards and thanks,
>
> The Authors

---

### Decision · Program_Chairs · 2025-09-17

**Decision:**

Accept (poster)

**Comment:**

The paper proposes a reinforcement learning framework that enhances vision-language models for retrieval-augmented generation (RAG).

Reviewers initially raised concerns about novelty, clarity of contributions, reward design, and missing ablations or comparisons. These issues were largely addressed in the rebuttal through additional experiments (reward-model swaps, sensitivity analysis, new ablations), extended comparisons with Vision-R1, Perception-R1, and VLM-R1, detailed failure case taxonomy, and clarifications of figures and design choices.


The strengths of the paper are consistently highlighted. Reviewer FHAe praised the well-motivated visual action space and clear empirical gains, Reviewer jJNP recognized significant performance improvements and effective use of visual perception actions, Reviewer 8qoA emphasized the novelty of visual perception actions, and Reviewer hXKu commended the integration of retrieval efficiency reward and extensive experiments showing strong performance improvements. All reviewers positively support this paper. Therefore, the AC recommend acceptance.